# Effect of Six-Week Speed Endurance Training on Peripheral Fatigue

**DOI:** 10.3390/ijerph191710841

**Published:** 2022-08-31

**Authors:** Blaz Jereb, Vojko Strojnik

**Affiliations:** Faculty of Sport, University of Ljubljana, 1000 Ljubljana, Slovenia

**Keywords:** training, muscle fatigue, peripheral fatigue, Wingate test

## Abstract

(1) Speed endurance training (inducing a high blood lactate concentration) delays excitation–contraction coupling impairment, thus providing more space for high-frequency fatigue to occur in the early stage of maximal concentric actions. This study aimed to test the hypothesis that the maintenance type of speed endurance training may shift peripheral fatigue from low-frequency to high-frequency fatigue after the 15 s long Wingate test. (2) Six students of physical education performed the corresponding training for six weeks. Before and after this period, they were tested for low- and high-frequency fatigue after the 15 s long Wingate test; additionally, their blood lactate concentrations, maximal cycling power, work, fatigue index, and muscle twitch responses were also tested. (3) The training increased the maximal cycling power and work (*p* < 0.001 and *p* < 0.01, respectively) with minor changes in the mean fatigue index and blood lactate concentration (both *p* > 0.05). Low-frequency dominant fatigue before the training showed a trend toward high-frequency dominant fatigue after the training (*p* > 0.05). (4) The results showed that the 15 s Wingate test failed to induce significant high-frequency fatigue. Even though it displayed a substantial fatigue index, the changes in favor of high-frequency fatigue were too small to be relevant.

## 1. Introduction

Muscle fatigue is associated with the inability to maintain the required force or power for a given activity [1]. The cause of muscle fatigue is thought to be an impairment of the contractile mechanism of the muscle fibers. This may be due to the accumulation of metabolic waste in the working muscles or to the depletion of energy stores in the muscle. Several mechanisms have been proposed, including the accumulation of intracellular lactate and hydrogen ions, ionic changes in the action potential, the failure of SR Ca^2+^ release, and the decreased calcium sensitivity of myofibrillar proteins [2,3,4]. Several fatigue mechanisms may be involved simultaneously and may differ depending on the motor task [5,6].

Muscle fatigue, also termed peripheral fatigue, can be studied by the electrical stimulation of muscles. Peripheral fatigue is divided into high-frequency and low-frequency fatigue, where high-frequency fatigue is related to the impairment of muscle action potential propagation and low-frequency fatigue is related to the impairment of activation–contraction coupling [7]. After maximal and near maximal exercise, both types of peripheral fatigue can occur. However, their dominance may differ depending on the type, intensity, and duration of the muscle contractions. In concentric contractions, regardless of the intensity and duration, low-frequency fatigue is predominant [8,9,10,11,12], whereas peripheral fatigue in a stretch–shortening cycle (SSC) may differ depending on the intensity and duration of the contractions. Maximal short SSC actions result in high-frequency fatigue [13,14], but when prolonged or performed at a submaximal intensity, low-frequency fatigue dominates [11,15,16].

The differences in fatigue responses may have implications for a training mode that reduces the occurrence of fatigue during maximal and near-maximal actions. Traditionally, training loads that produce high blood lactate concentrations (and consequently induce low-frequency fatigue) have been used mainly to prevent or delay a decline in performance at the maximal or near-maximal performance levels. On the other hand, a simple, reliable, and valid test for peripheral fatigue during maximal and near-maximal workouts is of great interest. In this context, maximal SSC exercises such as hopping or consecutive drop jumps could be used. The problem is that after one minute of a maximal SSC exercise the mechanical parameters do not change significantly [10]. Therefore, mechanical analysis of SSC jumping on a force plate or other equipment will not provide useful (if any) results in terms of fatigue.

The present study aimed to test the effect of maintenance-type speed endurance training [17] to challenge the anaerobic glycolytic system and shift fatigue from low to high frequency fatigue after a short maximal concentric load. The idea was based on the assumption that during prolonged maximally performed SSC activities, the initially observed high-frequency fatigue shifts to low-frequency fatigue and that this can also be observed during maximal concentric actions. Thus, if the impairment of excitation–contraction coupling can be delayed during brief maximal concentric actions, high-frequency fatigue should predominate. As the previous studies involved anaerobically untrained subjects, we decided to train the subjects accordingly. The Wingate test on a stationary bicycle was chosen because it measures peak anaerobic power and capacity [18], is a reliable test [19], and is considered the “gold standard” [20,21] in this area. As known from previous studies, both types of peripheral fatigue have occurred, with low-frequency fatigue predominating after the 15 s Wingate test [22]. Therefore, it is expected that improved excitation–contraction coupling after such training would reduce the occurrence of low-frequency fatigue and reveal high-frequency fatigue [23]. Since a significant decrease in cycling performance can be observed after only 15 s of the Wingate test [24], this opens the possibility of relating a mechanical fatigue index to high-frequency fatigue in trained subjects.

## 2. Materials and Methods

### 2.1. The Subjects

Six physical education students voluntarily participated in the study (age = 22.3 ± 2.5 years, height = 175 ± 1.5 cm, body weight = 69.3 ± 4.5 kg, and BMI = 22.5 ± 0.9). Informed consent was obtained from all subjects participating in the study, who stated that they were aware of the requirements, objectives, and potential risks. The study was conducted in accordance with the guidelines of the Declaration of Helsinki and was approved by the National Medical Ethics Committee (protocol code 22/9/96).

### 2.2. Experimental Design

The study was a single-group controlled trial with two assessment periods (pre- and post-intervention). Subjects were familiarized with the measurement protocol. Before the study, they had a trial session and a training session. Most of the participants had already taken part in similar studies at the Faculty of Sports in Ljubljana and were therefore informed about testing procedures. Before warming up, stimulation electrodes were applied to the subjects’ vastus lateralis (VL), and the device for measuring isometric knee torque was adapted to the size of the individual subject and to the ergometer load and wheel geometry for the Wingate test. The amplitudes of electrical stimulation (ES) for torque measurement at twitch, 20 HZ, and 100 Hz were established. A blood sample was taken one minute before the start of the warm-up period. The warm-up procedure consisted of stepping on a 20-cm bench at a step frequency of 0.5 Hz for 10 min and changing the pushing leg every minute. The step frequency was set with a metronome. After the warm-up, measurements were recorded with a two-and-a-half-minute delay due to blood sampling, followed by the ES tests. The timing of the ES tests was computer-controlled and kept constant during all measurements (three single twitches, 10 s delay, 20 Hz, and 100 Hz ES). The Wingate test was performed seven minutes after completion of the warm-up. The second measurements took place 60 s after the end of the Wingate test. ES tests were initially performed in the same manner as before. Blood samples were collected at 1 (after ES), 3, 5, 7, and 10 min after the Wingate test.

### 2.3. Exercise

The Wingate test was performed in a laboratory setting on a bicycle ergometer (Monark model 818E). The height of the seat was adjusted so that the knee was slightly flexed at the lowest pedal position. Body inclination during pedaling was kept constant by extended arms. Subjects were required to cycle at maximum intensity for 15 s against a resistance of 7.5% of body weight. Cycling started with a flying start so that the subject reached 60 turns per minute just before the actual start of the exercise. The bicycle was secured to the ground to ensure a stable pedaling motion. The power–time curve was recorded online on a computer screen in front of the subjects. Subjects were also cheered on by staff. The entire cycling protocol was performed according to the manufacturer’s instructions [25]. The maximum power, minimum power, average power, and work during the 15 s exercise were calculated based on a sensor that measured the speed of the wheel and considered the given resistance (SMI, Model 1000, St. Cloud, MN, USA). Power was normalized to body mass. A fatigue index was calculated as the ratio between the maximum power (P_max_) and the minimum power (P_min_) during the 15 sec cycle:(1)$$\mathrm{FI}=100-\left(\frac{P_{\min}}{P_{\max}}\cdot 100\right)$$

### 2.4. Anaerobic Training

The training period lasted 6 weeks with two or three training units (TU) per week (Table 1). The first four training units were implemented for adaptation to the heavy effort. Later, the training units were tailored to each individual as this was conducive to obtaining the closest to the desired effects of the exercise process. Before the training unit, subjects warmed up for 10 min at a load of 2.5% of body weight and 70 turns per minute. Ten minutes after warming up, they were tested with a 10-s Wingate test to determine their maximum power (100%). The intensity of the exercise was then adjusted to 65% of the maximum. The duration of the sets was always the same, except for the last set, which depended on the occurrence of fatigue. The occurrence of fatigue meant the beginning of a drop in the performance curve below the prescribed level, which the subjects followed on the screen in front of them and could not maintain despite applying maximum effort. If fatigue occurred in the last third of the set, the subject had to continue until the end of the set and then perform another set. If fatigue occurred before the last third of the set, the subject performed the set to the end with maximum effort, thus ending the training unit. If the subject completed more than six sets of the training session, the intensity was increased by five percent on the next set. The intensity was reduced by five percent if the subject completed fewer than four sets.

### 2.5. Knee Torque Measurements

Knee torque was measured on an isometric knee torque measurement device at a knee joint angle of 45°. During the measurements, the subject lay dorsally on the table with the hips locked and the lumbar spine supported to prevent movement of the pelvis. The distal part of the right shank was fixed to the lever and the knee axis was aligned with the axis of the measurement device. The self-adhesive 5 × 5 cm electrodes (Axelgaard, Falbrook CA) were placed over the vastus lateralis muscle (VL): anode over the distal part of the muscle belly and cathode over the middle part of the muscle belly. A computer-controlled constant current stimulator was used for muscle stimulation. Symmetrical quadratic biphasic pulses were used in all cases. The knee torque signal was recorded at 1000 Hz using DasyLab 7.0 (National Instruments). The timing of the electrical stimulation was controlled by computer. Signal analysis was performed using dedicated software.

#### 2.5.1. Twitch

A single supramaximal electrical pulse with 0.3 ms duration was delivered to relax the muscle VL. The current used to elicit a maximal twitch was determined by increasing the stimulation current until no further increase in torque was observed despite further increment in current. The current at maximum twitch torque was then increased by a further 50% to ensure a supramaximal stimulus. Maximum knee torque (FTW), contraction time (CT), and half relaxation time (HRT) were analyzed. CT began at 5% of FTW above the relaxation curve. HRT was defined as the time from FTW to the instant when the knee torque fell below half of FTW. Three consecutive twitches were measured with a delay of 1 s.

#### 2.5.2. Torque during 20 Hz and 100 Hz Electrical Stimulation

The relaxed muscle VL was consecutively electrically stimulated at a frequency of 20 Hz for 0.8 s and at 100 Hz for 0.8 s. The duration of a single electrical pulse was 0.3 ms. The amplitude of electrical stimulation was three times the motor threshold. The motor threshold was defined as the smallest electrical current that elicited the first visually perceptible VL muscle response during 100 Hz stimulation. The mean torque during the last 0.2 s of electrical stimulation at 20 Hz (T20) and 100 Hz (T100) was calculated.

### 2.6. Blood Analysis

Blood samples (20 µL) for blood lactate concentration ([La^−^]_b_) analysis were taken from the hyperemic ear. Blood lactate concentration was measured with the Kontron 640 Lactate Analyzer (Kontron, Vienna, Austria) immediately after the sample has been taken, i.e., after warm-up and 1, 3, 5, 7, and 10 min after the Wingate test.

### 2.7. Statistical Methods

The differences between the results measured before and after training were tested with a paired Student’s t-test. Statistical significance was accepted at 5% alpha error. The SPSS statistical package (SPSS Inc., Chicago, IL, USA) was used for analysis. The sample size was determined with an estimated effect size between 1 and 1.5, which would yield a sample size between 6 and 10 for the paired t-test with an error probability alfa = 0.05 and power = 0.80. We increased the sample size to 12 to account for possible dropouts (estimated to be 1 or 2 subjects). Unfortunately, some of the participants stopped participating in the study for various reasons. The actual number of subjects who completed the study according to the study protocol was 6, which is the minimum size required to reject the null hypothesis.

## 3. Results

The results for the mechanical, blood lactate, and twitch parameters are shown in Table 2.

### 3.1. Mechanical Parameters

Training statistically significantly increased the mean maximum power by 9.2% (*p* < 0.001) from 10.61 ± 2.38 W/kg to 11.59 ± 2.24 W/kg (mean and SD). Similarly, the mean work increased in a statistically significant manner by 7.2% (*p* < 0.01) from 9.99 ± 1.13 kJ to 10.71 ± 1.02 kJ. However, the change in the mean fatigue index (FI) from 19.61 ± 3.42% to 23.06 ± 3.08% was not statistically significant (*p* > 0.05).

### 3.2. Blood Lactate

Before training, the highest blood lactate concentration was measured in the third minute after the Wingate test (6.15 mM/µL). After training, the blood lactate concentration behaved almost identically to that before training (*p* > 0.05).

### 3.3. Twitch

From before to after training, the mean FTW before the 15 s Wingate test increased from 22.48 ± 13.96 Nm to 28.66 ± 10.36 Nm. However, the change was not statistically significant (*p* > 0.05). Before training, the mean decrease in the FTW by the 15 s Wingate test was 8.52 Nm and after training was 8.06 Nm. The difference was not statistically significant (*p* > 0.05). A similar behavior was observed for the mean value CT. It increased from 71.66 ± 8.93 ms to 80.16 ± 7.38 ms before and after training, respectively. The change was not statistically significant (*p* > 0.05). The mean changes after the 15 s Wingate test were minimal (less than 2 ms) and not statistically significant (*p* > 0.05). A similar behavior was observed with the HRT.

### 3.4. Torque during 20 Hz and 100 Hz Electrical Stimulation

The mean T20 measured before the 15 s Wingate test did not change in a statistically significant fashion after training (from 19.14 ± 7.6 Nm to 16.46 ± 7.66 Nm, *p* > 0.05) (Figure 1). The changes in T20 after the Wingate test were statistically significant before training (mean T20 decreased by 3.11 Nm, *p* < 0.01) but not after training (mean T20 increased by 0.29 Nm, *p* > 0.05). The mean T100 measured before the 15 s Wingate test did not change statistically significantly after training (from 31.00 ± 13.22 Nm to 29.54 ± 14.30 Nm, *p* > 0.05). The changes in T100 after the Wingate test were statistically significant before training (mean T100 decreased by 2.49 Nm, *p* < 0.01) but not after training (mean T100 decreased by 0.93 Nm, *p* > 0.05). The mean T100/T20 ratio changed by 11.78%, from 1.08 before training to 0.96 after training.

## 4. Discussion

The results show that the maintenance form of speed endurance training, which aims to challenge the anaerobic glycolytic system, increases maximal power as well as work during the 15 s Wingate test. Although the fatigue index increased after the training, the difference was not statistically significant. The changes in the blood lactate concentration after the Wingate test were almost identical before and after training. The T20 after the Wingate test was decreased before training but not after training. Similarly, the T100 for the Wingate test decreased before training, whereas no significant changes were observed after training. The T100/T20 index indicated a shift from low-frequency-dominated peripheral fatigue toward high-frequency fatigue, although not a statistically significant one.

Anaerobic training was effective at increasing anaerobic strength and work. Since there were no changes in the blood lactate concentration and fatigue index, it is reasonable to assume that the training improved muscle activation, anaerobic alactic energy systems, and buffer systems. In addition, muscle hypertrophy, which increases the absolute ATP and CrP [26], could also explain the results obtained. However, this was less likely due to the short training duration [27] and training mode [28], which did not support substantial muscle hypertrophy.

The lack of statistically significant differences between the changes in the FTW before and after training might suggest that the fatigue mechanisms did not change. However, the subjects performed more work at an increased maximal power. The decrease in the FTW before and after training might have been influenced by a decrease in the intracellular pH. The increase in the buffer capacity [29] may have reduced the decrease in the intracellular pH so that excitation–contraction coupling functioned better after training than before training. In rats, Viru [30] found that the Ca^2+^ accumulation rate in the sarcoplasmic reticulum increased after ten weeks of sprinting, interval aerobic running, and explosive strength training. Therefore, a slight decrease in FTW after the training process could be due to a decreased capacity of the buffer systems or an increased Ca^2+^ content in the sarcoplasmic reticulum and terminal cistern.

Short maximal exercises such as the 15 s Wingate test rely largely on energy derived from phosphocreatine hydrolysis and only partially from anaerobic glycolysis. This leads to an intramuscular accumulation of metabolic byproducts (e.g., Pi, K^+^, Na^+^, H^+^, Ca^2+^, and Cl^−^) that impair Ca^2+^ release from the sarcoplasmic reticulum (SR) [31,32] or the decreased Ca^2+^ sensitivity of myofibrils [11]. Such accumulation could be an explanation for the decreased T20 observed after the Wingate test before the training [33]. Since the T20 did not change after training, the training regime appeared to be able to preserve the Ca^2+^ release from SR and Ca^2+^ binding to troponin during the first 15 s of the Wingate test. Since the electrical stimulation of 20 Hz is not sufficient for complete tetanus, changes in twitch times could affect the T20. In the present study, this was not very likely because the changes in the CT and HRT were minor and not statistically significant. The changes related to low-fatigue were in line with our expectations and opened the possibility of more pronounced high-frequency fatigue.

However, the decrease in the T100 after training was not statistically significant. This could be due to an improvement in the propagation of muscle action potentials [34]. The speed of the propagation of muscle action potentials depends on the number of Na-K pumps [35]. The critical sites for action potential propagation are the t-tubules, where the blockage of propagation can occur [35,36], regardless of the decreased pH [37]. The greater decrease in the T100 after the training could be prevented by a greater number of Na-K pumps, as their number may increase during high-intensity training [17,38]. Consequently, the value of the T100/T20 was shifted toward high-frequency fatigue.

In the past, such a shift in the type of peripheral fatigue was observed only after short maximal stretch–shortening cycles. It should be noted that there are still significant differences between maximal concentric and SSC actions with respect to inducing high-frequency fatigue. They are evident in the duration of the load and the magnitude of the high-frequency fatigue induced. At maximal SSC actions, high-frequency fatigue was observed after 60 s or an even longer duration [13,14,39]. As shown in the present study, the maximum duration of concentric action was much shorter—15 s. An important difference also exists in the magnitude of high-frequency fatigue, which is much more pronounced in SSC actions. Consequently, it seems that the impairment of muscle action potential propagation along the t-tubules is not a major fatigue problem not only for short maximal but for all concentric actions.

## 5. Conclusions

The results obtained may have practical relevance. One of these relevancies concerns training. It seems reasonable to assume that speed endurance training affects both types of peripheral fatigue, as shown in the present study. However, the main effect is the avoidance of low-frequency fatigue. Since significant high-frequency fatigue occurs during maximal or near-maximal SSC actions (e.g., ski slalom) with little change in blood lactate concentration [14], speed endurance training may not be the optimal solution for preventing fatigue in such sports. Whether SSC training would better serve against high-frequency fatigue is still to be determined. Another possible practical application would be to use the 15 s Wingate test to determine high-frequency fatigue. The results show that the short Wingate test does not induce significant high-frequency fatigue. Although it showed a significant fatigue index, the changes in favor of high-frequency fatigue were too small to be relevant. Therefore, the 15 s Wingate test is not a good candidate for high-frequency fatigue testing.

The results obtained confirmed our hypothesis that the maintenance type of speed endurance training may shift peripheral fatigue from low-frequency to high-frequency fatigue during maximal concentric actions of short duration in strength endurance training. However, because of the small and statistically non-significant changes and the small number of subjects, the results should be considered with caution.

The present study has some limitations that should be considered. The number of subjects was small, which limits the generalizability of the results. The significant proportion of drop-outs was mainly due to the very demanding training regime associated with maximal effort and pain, as these increase the perceived fatigability [40]. Nevertheless, the results appeared to be consistent with previous studies and expectations. The training level of the subjects was not as high as it might be in elite athletes. This leaves some room for a potentially larger shift toward high-frequency fatigue [41,42]. If such a shift would be substantial, it would potentially qualify the 15 s Wingate test for testing high-frequency fatigue. However, the validity of such a test would still be questionable because it depends on the training status of the person taking the measurement.

## Figures and Tables

**Figure 1 ijerph-19-10841-f001:**
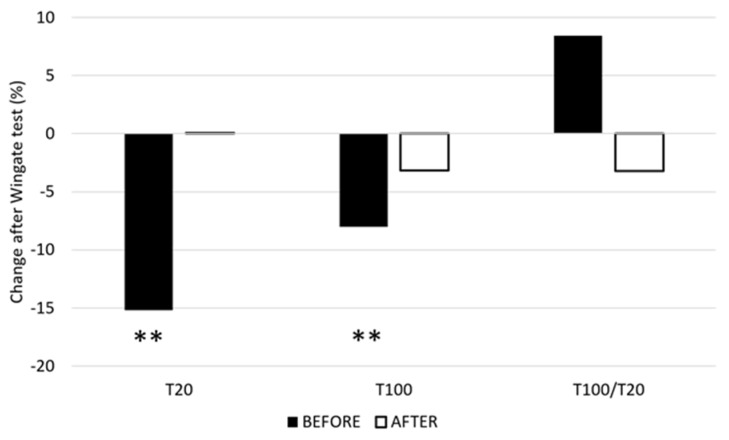
Relative mean changes of T20 and T100 before and after the training. T20—torque during 20 Hz ES, T100—torque during 100 Hz ES, and T100/20—ratio between T100 and T20. **—*p* ≤ 0.01.

**Table 1 ijerph-19-10841-t001:** Presentation of the training load.

Week	TU/Week	Sets	Duration (s)	Rest	Intensity
1.	2	3	35 s	1 min	60%
2.	2	4	35 s	1 min	60–70%
3.	3	4–6	35 s	1 min	60–70%
4.	2	4–6	35 s	1 min	60–70%
5.	3	4–6	35 s	1 min	60–70%
6.	2	4–6	35 s	1 min	60–70%

**Table 2 ijerph-19-10841-t002:** Mechanical parameters, Twitch parameters, and Blood lactate.

	BEFORE		AFTER		*p*
Max Power (W/kg)	10.62	2.38	11.59	2.24	**
Work (kJ)	9.99	1.13	10.71	1.02	***
FI (%)	19.61	3.42	23.06	3.08	
FTW (Nm)	22.48	13.96	28.66	10.36	
CT (ms)	71.66	8.93	80.16	7.38	
HRT (ms)	40.16	12.46	51.16	8.65	
[La^−^]_b_ (mM/µL)	6.15	0.75	5.86	0.32	

Max Power—mean maximum power normalized to body mass, Work (kJ)—mean work, FI (%)—fatigue index, FTW—maximal knee torque, CT—contraction time, HRT—half relaxation time, [La^−^]_b_—blood lactate concentration, and *p*—statistical significance. ** *p* ≤ 0.01; *** *p* ≤ 0.001.

## Data Availability

The raw data supporting the conclusions of this article will be made available by the authors, without undue reservation.

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
