# Peer review of "Effect of Six-Week Speed Endurance Training on Peripheral Fatigue"

_ijerph, 2022, doi:10.3390/ijerph191710841_

Round 1
Reviewer 1 Report
Dear authors,
General Comment
Your research has focused on peripheral fatigue, which has not given much consideration to its subject matter. In particular, evaluating your research with a 6-week exercise program further increases the value of your results. In this respect, your research has a high originality value and contributes to the field of sports sciences. In general, the references you used in your research are very old and you should discuss your research findings with current references, using more recent references. Although not directly related to your subject, there are many current references in the literature similar to the subject of your research.
As conclusion, I will suggest some important corrections to increase the readability of your research. After these corrections, I think your research is fit for publication in ijerph.
Abstract
-Please add background to your abstract before the purpose sentence.
-Please indicate all measurements you have made in this section.
Introduction
-The introduction is generally adequate, but almost all the references to your work are very old, please try to revise this section by providing up-to-date sources (main deficiency of your old reference use research)
Materials and methods
You used only six subjects in your study. Is this number of subjects sufficient for the validity of the study? Please prove whether your number of subjects is sufficient with gpower etc power analysis
Please expand your experimental design (for example, which tests were done in what order? How did you rest between tests (days or hours)? Did you perform a familization session on the subjects before the tests? You can write all this down or present it as a diagram.
Results
The results you present in the findings are compatible with your research purpose, but giving all the findings in tables will help the readers to understand more quickly.
Please present the statistical analysis results of the parameters such as mechancal parameters, blood lactate etc. in tables or figures.
Discussion
-Please revise your discussion using more recent references instead of outdated references (this is a suggestion).
-Also, please include the limitations of your research at the end of this section.
Author Response
Dear Sir,
Thank you for your comments and for the opportunity to submit a revised manuscript. We appreciate all feedbacks on our manuscript that was revised according to your comments. Point by point explanations are attached. The content corrections are coloured within the manuscript. English revision was done, too.
We look forward to hearing from you.
Sincerely,
Dr. Blaz Jereb

Reviewer 2 Report
The manuscript is interesting, well written and points out relevant issues.
My comments:
· Line 74-75: The code of the Ethics Committee statement is missing.
· Line 97- 98: Suggestion: “A fatigue index was calculated as a ratio between the maximum power (Pmax) and the minimum power (Pmin), during the 15 s cycle:“
· Line 158: Why "code" at the end of the sentence?
· Line 168: Suggestion: “Before training, the…”
· Line 170: Suggestion: “…after training, behaved…”
· Line 201: Confirm the need for the word “Authors” at the beginning of the sentence
As an additional opinion: The authors do not explore the possible influence of increasing the duration of the Wingate test (30s vs 15s), which would be interesting. The expression of fatigue depends on the intensity but also on the duration of the exercise.
Congratulations on the study!
Author Response
Dear Sir,
Thank you for your comments. We appreciate the time and effort that you dedicated to all feedbacks on our manuscript. The manuscript was revised according to your comments. Point by point explanations are attached. The content corrections are coloured within the manuscript. English revision was done, too.
We look forward to hearing from you.
Sincerely,
Dr. Blaz Jereb

Reviewer 3 Report
In the paper by Blaž Jereb et al., titled “Effect of a 6-Week Speed Endurance Training on Peripheral Fatigue” the authors aimed to test hypothesis that maintenance type of speed endurance training (inducing high blood lactate concentration) may shift peripheral fatigue from low-frequency to high-frequency fatigue in maximal concentric actions of short duration.
The article is written on an intriguing and up-to-date topic of modern sport science and has a scientific interest, has merit although some minor issues could be addressed so to even improve the overall quality.
However, several issues need to be addressed:
1. Six students of physical education volunteered in the study (age = 22.3 ± 2.5 yr, height 70 = 175 ± 1.5 cm, body mass = 69.3 ± 4.5 kg). This is all the evaluation done of the students. BMI or any data on body composition assessment?
2. Heart Rate? no data on this nor variability of cardiac frequency.
3. To analyze indicate fatigue would not be appropriate to analyze marker Creatine Phosphokinase? (CPK)
4. Please update the references, are old in there are several references related to the theme, more current (80-90s).
5. Six students? Is there no way to increase the sample?
6. How does the author classify the extent of the physical exercise protocol? Short, medium, or long-term?
Author Response
Dear Sir,
Thank you for your comments and for the opportunity to submit a revised manuscript. We appreciate all feedbacks on our manuscript. It was revised according to your comments. Point by point explanations are attached. The content corrections are coloured within the manuscript. English revision was done, too.
We look forward to hearing from you.
Sincerely,
Dr. Blaz Jereb

Round 2
Reviewer 1 Report
Dear Authors,
Congratulations. Your manuscript is ready for publication. Thank you for your efforts.
Best Regards